# Variability in locomotor dynamics reveals the critical role of feedback in task control

Ismail Uyanik[1,2,3]*, Shahin Sefati[4], Sarah A Stamper[4], Kyoung-A Cho[4], M Mert Ankarali[5], Eric S Fortune[3†], Noah J Cowan[2,4†]*

[1]Department of Electrical and Electronics Engineering, Hacettepe University, Ankara, Turkey; [2]Laboratory of Computational Sensing and Robotics, Johns Hopkins University, Baltimore, United States; [3]Department of Biological Sciences, New Jersey Institute of Technology, Newark, United States; [4]Department of Mechanical Engineering, Johns Hopkins University, Baltimore, United States; [5]Department of Electrical and Electronics Engineering, Middle East Technical University, Ankara, Turkey

**Abstract** Animals vary considerably in size, shape, and physiological features across individuals, but yet achieve remarkably similar behavioral performances. We examined how animals compensate for morphophysiological variation by measuring the system dynamics of individual knifefish (*Eigenmannia virescens*) in a refuge tracking task. Kinematic measurements of *Eigenmannia* were used to generate individualized estimates of each fish's locomotor plant and controller, revealing substantial variability between fish. To test the impact of this variability on behavioral performance, these models were used to perform simulated 'brain transplants'— computationally swapping controllers and plants between individuals. We found that simulated closed-loop performance was robust to mismatch between plant and controller. This suggests that animals rely on feedback rather than precisely tuned neural controllers to compensate for morphophysiological variability.

**\*For correspondence:**
uyanik@ee.hacettepe.edu.tr (IU);
ncowan@jhu.edu (NJC)

[†]These authors contributed equally to this work

**Competing interests:** The authors declare that no competing interests exist.

## Introduction

Animals routinely exhibit dramatic variations in morphophysiology between individuals but can nevertheless achieve similar performance in sensorimotor tasks (*Sponberg et al., 2015*; *Bullimore and Burn, 2006*). Further, individual animals can experience rapid changes in their own morphophysiological features, such as extreme weight changes that occur during and between bouts of feeding. For example, mosquitoes can consume more than their body weight (*Van Handel, 1965*) and hummingbirds can consume up to 20% of their body weight (*Hou et al., 2015*) in a single feeding. How neural control systems accommodate these changes is not known.

The behavioral performance of an individual animal is determined via an interplay between its 'controller' and 'plant' (*Kiemel et al., 2011*; *van der Kooij and Peterka, 2011*; *Cowan et al., 2014*; *Dickinson et al., 2000*; *Hedrick et al., 2009*). The plant typically includes musculoskeletal components that interact with the environment to generate movement (*Hedrick and Robinson, 2010*; *Sefati et al., 2013*; *Maladen et al., 2009*). The controller typically includes sensory systems and neural circuits used to process information to generate motor commands (*Cowan and Fortune, 2007*; *Kiemel et al., 2011*; *Lockhart and Ting, 2007*; *Roth et al., 2014*). From the perspective of control theory, one might expect the dynamics of the controller to be precisely tuned to the dynamics of the plant, resulting in an optimal control law that reduces variability in task performance (*Todorov, 2004*; *Franklin and Wolpert, 2011*; *Bays and Wolpert, 2007*). Were this the case,

**eLife digest** People come in different shapes and sizes, but most will perform similarly well if asked to complete a task requiring fine manual dexterity – such as holding a pen or picking up a single grape. How can different individuals, with different sized hands and muscles, produce such similar movements? One explanation is that an individual's brain and nervous system become precisely tuned to mechanics of the body's muscles and skeleton. An alternative explanation is that brain and nervous system use a more "robust" control policy that can compensate for differences in the body by relying on feedback from the senses to guide the movements.

To distinguish between these two explanations, Uyanik et al. turned to weakly electric freshwater fish known as glass knifefish. These fish seek refuge within root systems, reed grass and among other objects in the water. They swim backwards and forwards to stay hidden despite constantly changing currents. Each fish shuttles back and forth by moving a long ribbon-like fin on the underside of its body. Uyanik et al. measured the movements of the ribbon fin under controlled conditions in the laboratory, and then used the data to create computer models of the brain and body of each fish. The models of each fish's brain and body were quite different.

To study how the brain interacts with the body, Uyanik et al. then conducted experiments reminiscent of those described in the story of Frankenstein and transplanted the brain from each computer model into the body of different model fish. These "brain swaps" had almost no effect on the model's simulated swimming behavior. Instead, these "Frankenfish" used sensory feedback to compensate for any mismatch between their brain and body.

This suggests that, for some behaviors, an animal's brain does not need to be precisely tuned to the specific characteristics of its body. Instead, robust control of movement relies on many seemingly redundant systems that provide sensory feedback. This has implications for the field of robotics. It further suggests that when designing robots, engineers should prioritize enabling the robots to use sensory feedback to cope with unexpected events, a well-known idea in control engineering.

variations across individuals in morphophysiological features of their plants should manifest in commensurate differences in each animal's controller. Alternatively, the central nervous system (CNS) may be implementing robust feedback control that attenuates morphophysiological variability at the behavioral level without the need for precise tuning.

Investigating these relationships requires separate estimates for plants and controllers. However, the classical input–output system identification of behavioral tasks—using only the sensory input and the behavioral output—is limited to generating closed-loop control models of behavioral responses. Data-driven system identification of the underlying neural controllers or locomotor plants requires additional observations such as a measurement of the control output. Electromyograms (EMGs) are the most commonly used proxy for the output of the neural controller. EMGs allow separate data-driven estimates of the controller and plant (*Kiemel et al., 2011*; *van der Kooij and Peterka, 2011*) but require understanding the coordination strategy across multiple groups of muscles that interact in fundamentally nonlinear ways (*Ting and Macpherson, 2005*).

We studied refuge tracking in a species of weakly electric fish *Eigenmannia virescens* (*Figure 1A*), a system that permits identification of input–output dynamics as well as the locomotor plant via behavioral observations alone. Like an 'aquatic hummingbird', *Eigenmannia* precisely hover in place, making rapid and nuanced adjustments to its position in response to the movement of the refuge in which it is hiding (*Rose and Canfield, 1993*; *Roth et al., 2011*; *Uyanik et al., 2019b*; *Figure 1—video 1*). During refuge tracking, individual *Eigenmannia* generate fore-aft thrust forces using undulatory motions of their ventral ribbon fin. Undulations are initiated at the rostral and caudal ends of the fin resulting in counter propagating waves that travel towards each other (*Sefati et al., 2013*; *Ruiz-Torres et al., 2013*). The two traveling waves meet at a position along the ribbon fin known as the nodal point (*Figure 1—video 2*). In a task in which the fish maintains position in a stationary refuge, *Eigenmannia* shift the rostrocaudal position of the nodal point as a function of steady-state swimming speed (*Sefati et al., 2013*; *Figure 1—figure supplement 1*), providing a behavioral proxy for the controller's output, without reliance on EMGs.

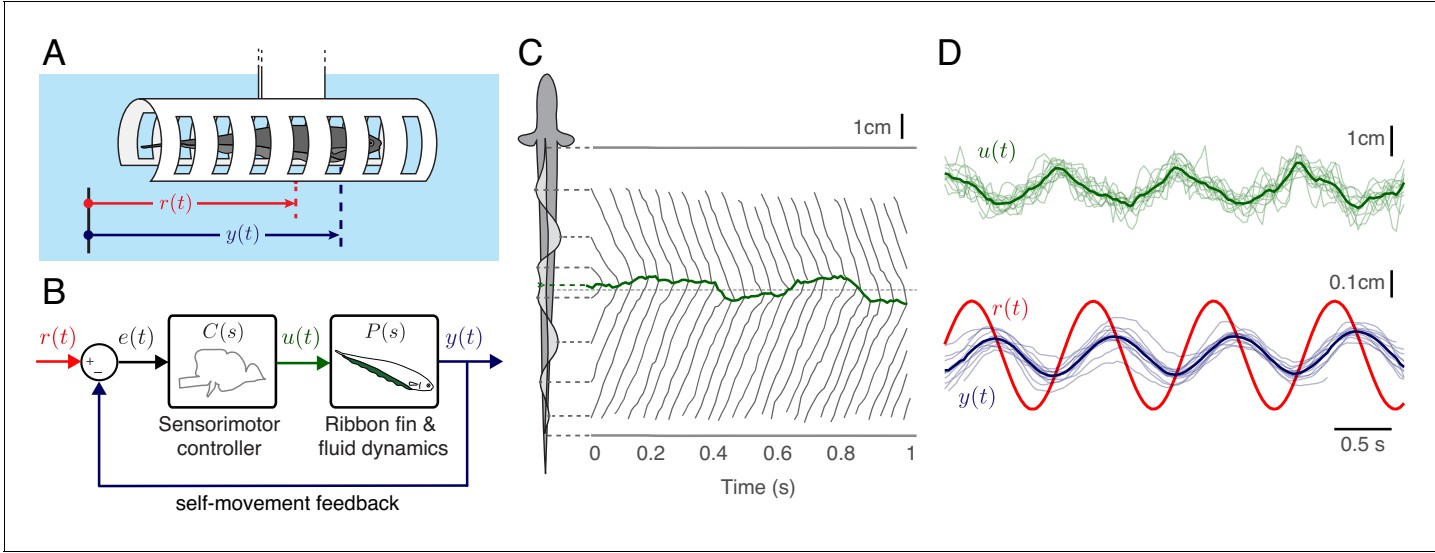

**Figure 1.** Experimental analysis of refuge tracking behavior. (**A**) *Eigenmannia virescens* maintains its position, $y(t)$, in a moving refuge. $r(t)$ is the position of the moving refuge. (**B**) A feedback control system model for refuge tracking. The feedback controller $C(s)$ maps the error signal, $e(t)$, to a control signal, $u(t)$. The plant $P(s)$, in turn, generates fore-aft movements $y(t)$. We used measurements of the nodal point as $u(t)$ to estimate $P(s)$ in each fish, which was subsequently used to infer $C(s)$. (**C**) *Eigenmannia virescens* partitions its ribbon fin into two counter-propagating waves that generate opposing forces (*Sefati et al., 2013*; *Ruiz-Torres et al., 2013*). The rostral-to-caudal and caudal-to-rostral waves collide at the nodal point (green line). Movements of the nodal point serve as a proxy for the control signal, $u(t)$. Gray lines indicate the positions of the peaks and troughs of the traveling waves over time. In this trial, the refuge was following a sinusoidal trajectory at 2.05 Hz. (**D**) Movement of the nodal point $u(t)$, position of the fish $y(t)$, and position of the refuge $r(t)$ of 13 trials at 0.95 Hz. Semi-transparent lines represent data from individual trials, green and blue lines are the means.

The online version of this article includes the following video and figure supplement(s) for figure 1:

**Figure supplement 1.** Steady-state position of the nodal point.
**Figure supplement 2.** Movement of the refuge, fish and nodal point.
**Figure 1—video 1.** A real-time video recorded at 30 fps from below of an *Eigenmannia virescens* tracking a moving refuge at 0.55 Hz.
https://elifesciences.org/articles/51219#fig1video1
**Figure 1—video 2.** A high-speed video recorded at 100 fps from below an *Eigenmannia virescens* while tracking a moving refuge at 2.05 Hz.
https://elifesciences.org/articles/51219#fig1video2

We measured tracking performance and moment-to-moment position of the nodal point in three fish during a refuge tracking task. Despite the fact that tracking performance of the three fish were similar, there were nevertheless large variations in the movement of the nodal point, reflecting morphophysiological differences between individuals. We used computational techniques, specifically data-driven system identification of feedback control models, to explore how neural control systems cope with individual variability in locomotor dynamics.

## Results

We measured the performance of three fish in a refuge tracking task by comparing the position of the refuge $r(t)$ with the position of the fish $y(t)$ (**Figure 1A**). These measurements are used in a feedback control model in which 'error signal' $e(t)$ is defined as the difference between $r(t)$ and $y(t)$ (**Figure 1B**). To separately estimate the controller $C(s)$ and the plant $P(s)$, a measurement of the control signal $u(t)$ is necessary. In *Eigenmannia*, the position of the nodal point can be measured over time (**Figure 1C**, **Figure 1—video 2**). During refuge tracking behavior, we observed that the position of the nodal point appears to have a linear relationship to $y(t)$ making it a candidate readout of $u(t)$ (**Figure 1D**). Using $u(t)$ and $y(t)$, we estimated $P(s)$ for each fish and calculated their corresponding controllers $C(s)$. We used $C(s)$ and $P(s)$ of each fish to computationally manipulate the interplay between the controller, plant and the sensory feedback.

## Estimating a data-driven plant model

We estimated a data-driven model for the plant dynamics $P(s)$ of each fish. For the purpose of visualizing the plant models graphically, it is useful to treat the plant, $P(s)$, as a filter through which motor commands are processed. At a given frequency, $\omega$, the filter's behavior can be represented as a complex number, called a phasor, that is obtained by evaluating the transfer function $P(s)$ at that frequency, namely $P(j\omega)$, where $j = \sqrt{-1}$.

The locus of phasors as a function of frequency is called the frequency response function. We estimated the frequency responses of the locomotor dynamics of each fish using the position of the fish $y(t)$ and its nodal point $u(t)$. To visualize the variability across plants, we used a Nyquist plot. On such a plot, the gain and phase of the response of the plant at a given frequency are represented in polar form, where the x axis is the real part and the y axis is the imaginary part—both dimensionless. The gain is the distance from the origin $0+j0$, and the phase is the angle measured counterclockwise from the real axis. Nyquist plots of each individual's estimated plant revealed substantial differences between locomotor plants of individual fish (*Figure 2A*). Despite these differences, the frequency responses shared a common structure: the locomotor dynamics of each fish acted as a low-pass filter on the movements of the nodal point. This common structure facilitated the application of parametric modeling, reducing the complexity of analysis while enabling computational manipulations of the model system. We used the physics-based parametric model of locomotor dynamics of *Eigenmannia* described by *Sefati et al. (2013)* for the plant (see Materials and methods for derivation):

$$P(s) = \frac{k}{ms^2 + bs} \tag{1}$$

Here, $m$, $k$, and $b$ represent mass, gain, and damping, respectively, and $s$ is complex frequency in Laplace domain (see, e.g. *Roth et al., 2014*). We estimated the parameters in the parametric plant model for each fish based on their individualized frequency responses via numerical optimization (see Materials and methods) (*Figure 2B*). Frequency responses for the estimated parametric models are illustrated in *Figure 2C* (see black lines in *Figure 2A* for corresponding Nyquist plots). Finally, we estimated the plant for a 'merged' fish, in which the data from the three fish were concatenated as a single fish. The differences in the frequency responses between individuals resulted in substantial differences (about twofold) in estimated model parameters (*Table 1*). Moreover, the merged fish has plant dynamics that differ from each of the individual fish (*Figure 2A*, bottom), highlighting the need to use individualized plants for the analysis of the control system of each fish.

## Examining the effects of feedback on behavioral variability

Despite the differences in plant dynamics, fish produced remarkably similar tracking performance, consistent with previously published reports (*Cowan and Fortune, 2007*; *Roth et al., 2011*). This behavioral robustness could be achieved via precise tuning between the controller and plant dynamics of each fish. Alternatively, the central nervous system (CNS) may be implementing robust feedback control without the need for precise tuning. To test these hypotheses, we built feedback control models that permit computational manipulation of the relationships between controller and plant. Specifically, we swapped the controllers and plants between fish using these computational models (*Figure 3*). If each fish required precise tuning for consistent behavior, we would expect to see increased variability for the swapped models. Alternatively, a robust feedback controller might be insensitive to mismatch between $C(s)$ and $P(s)$ pairs.

We used the second order model proposed by *Cowan and Fortune (2007)* to represent the input–output behavioral response of the fish:

$$G(s) = \frac{A\omega_n^2}{s^2 + 2\zeta\omega_n s + \omega_n^2} \tag{2}$$

We estimated the model parameters for each fish using the position of the fish $y(t)$ and the refuge $r(t)$. In other words, we generated individualized parametric transfer functions that capture the input–output behavioral performance of each fish. Parameters varied by about 15–20% (*Table 2*).

We investigated how the variability in plant dynamics (parameters varied by about twofold; *Table 1*) is mitigated at the level of behavior (parameters varied by 15–20%; *Table 2*). Specifically, we inferred a controller for each fish using models of their respective plant dynamics and input–

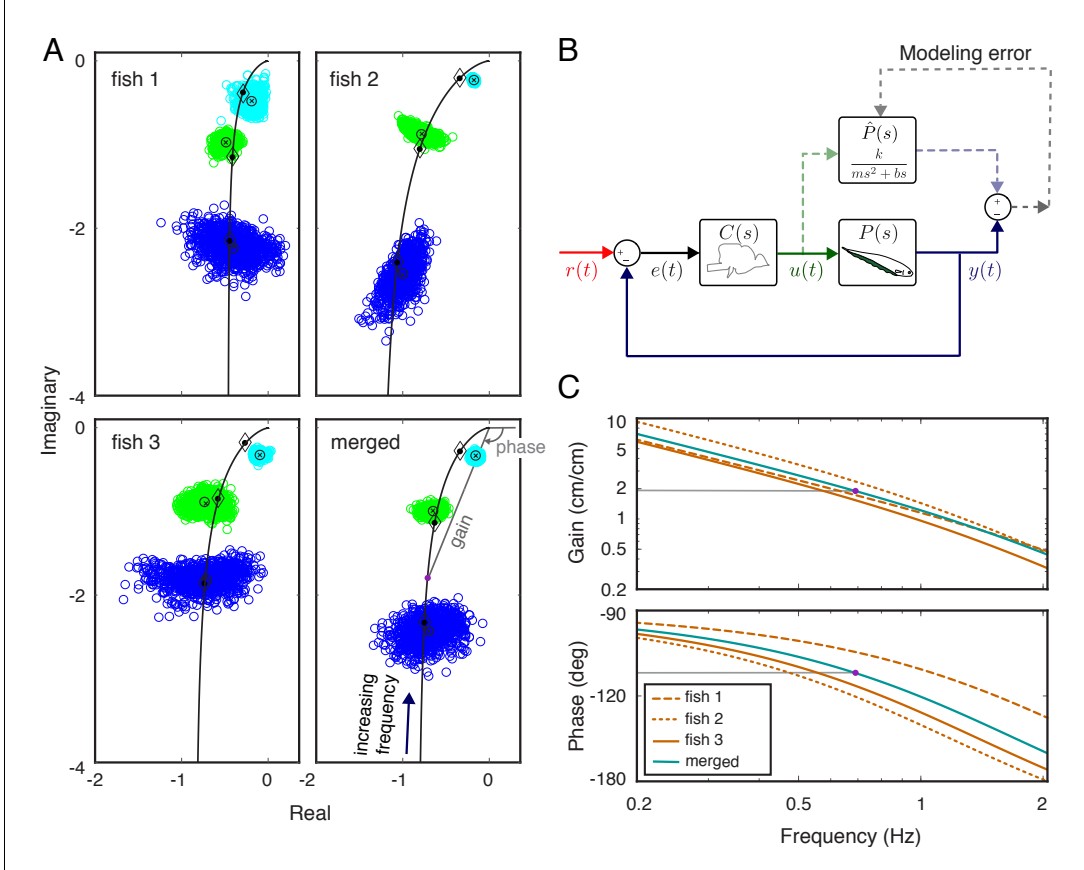

**Figure 2.** System identification of plant dynamics. (**A**) Nyquist plots of the estimated plant models for each individual fish and the merged fish. Blue, green and cyan point clouds correspond to bootstrapped data estimates for the low (0.55 Hz), medium (0.95 Hz) and high (2.05 Hz) frequencies, respectively. Black circles and cross marks represent the mean for measured and bootstrapped data, respectively. Black lines represent the response of the estimated model with diamonds indicating their values at the test frequencies. Each point on the black lines correspond to the complex-valued frequency response of the plant model at a specific frequency. For example, the purple dot on the lower-right panel corresponds to the merged-plant-model's response at *0.7 Hz*. The gain and phase associated with this response are shown on the associated Bode plots given in (**C**). (**B**) Reconciliation of physics-based and data-driven models. The solid lines represent the natural feedback control system used by the fish for refuge tracking. The dashed lines represent copies of signals used for parametric system identification. $\hat{P}(s)$ represents the parametric transfer function for the plant dynamics of the fish with 'unknown' system parameters. The parametric system identification estimates these parameters via minimizing the difference (modeling error) between the actual output of the fish $y(t)$ and the prediction of the model. (**C**) Gain and phase plots of the frequency response functions of the estimated parametric models for each fish and the merged fish corresponding to the black lines in (**A**).

output behavioral responses. Given the plant $P(s)$ and behavioral performance $G(s)$ of each individual fish, we can infer the controller for each fish using the following equation (see Materials and methods and *Cowan and Fortune, 2007*):

$$C(s) = \frac{G(s)}{(1 - G(s))P(s)} \tag{3}$$

**Table 1.** Estimated parameters of the plant model for each fish as well as the merged fish.

|  | *Fish 1* | *Fish 2* | *Fish 3* | *Merged* |
|---|---|---|---|---|
| $k\,(N/m)$ | 0.3228 | 0.2200 | 0.1622 | 0.2385 |
| $b\,(Ns/m)$ | 0.0417 | 0.0186 | 0.0218 | 0.0269 |
| $m\,(kg)$ | 0.0025 | 0.0025 | 0.0025 | 0.0025 |

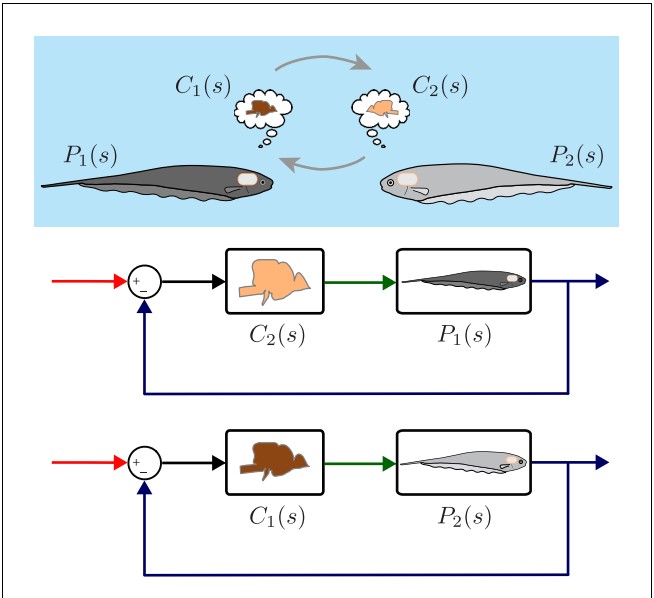

**Figure 3.** Computational brain swapping between fish. We computationally swapped the individually tailored controllers between fish. The feedback control diagrams illustrate computational brain swaps. The controller from the light gray fish, $C_2(s)$, is used to control the plant of the dark gray fish, $P_1(s)$, and vice versa.

Estimates of controllers and plants for individual fish allowed us to implement computational manipulations in the system models. Given a model of each fish's plant and controller, we computed the transfer function of the closed-loop system using each fish's own controller when matched with its own plant ('matched', *Figure 4A*), via the equation (see *Equation 11*, Materials and methods):

$$G_{\mathrm{matched},i}(s) = \frac{P_i(s)C_i(s)}{1 + P_i(s)C_i(s)}, \tag{4}$$

for $i = 1,2,3$. Then, to test the hypothesis that the animals rely on precise tuning between their plants and controllers, we substituted the controller of each fish with the plant dynamics of other fish (see 'swapped' in *Figure 4A*), that is a simulated brain transplant, namely

$$G_{\mathrm{swapped},i,j}(s) = \frac{P_i(s)C_j(s)}{1 + P_i(s)C_j(s)} \tag{5}$$

whereby each controller $j = 1,2,3$ is paired with another fish's plant $i \neq j$, for a total of 6 'swapped' cases. If the controllers and plants need to be co-tuned, then we would expect a significant increase in variability in the swapped models.

To quantify such changes in variability and to evaluate the fitness of a given computational model for explaining biological data, we defined two metrics termed 'model variability' and 'behavioral variability'. Model variability quantifies the variability of the complex-valued frequency responses of matched or swapped models across a range of frequencies. Behavioral variability, on the other

**Table 2.** Estimated parameters of the second order input-output model for each fish as well as the merged fish.

|  | Fish 1 | Fish 2 | Fish 3 | Merged |
|---|---|---|---|---|
| $A$ | 0.53 | 0.60 | 0.52 | 0.57 |
| $\xi$ | 0.58 | 0.55 | 0.52 | 0.55 |
| $\omega_n$ (*rad/s*) | 8.55 | 9.51 | 7.88 | 8.50 |

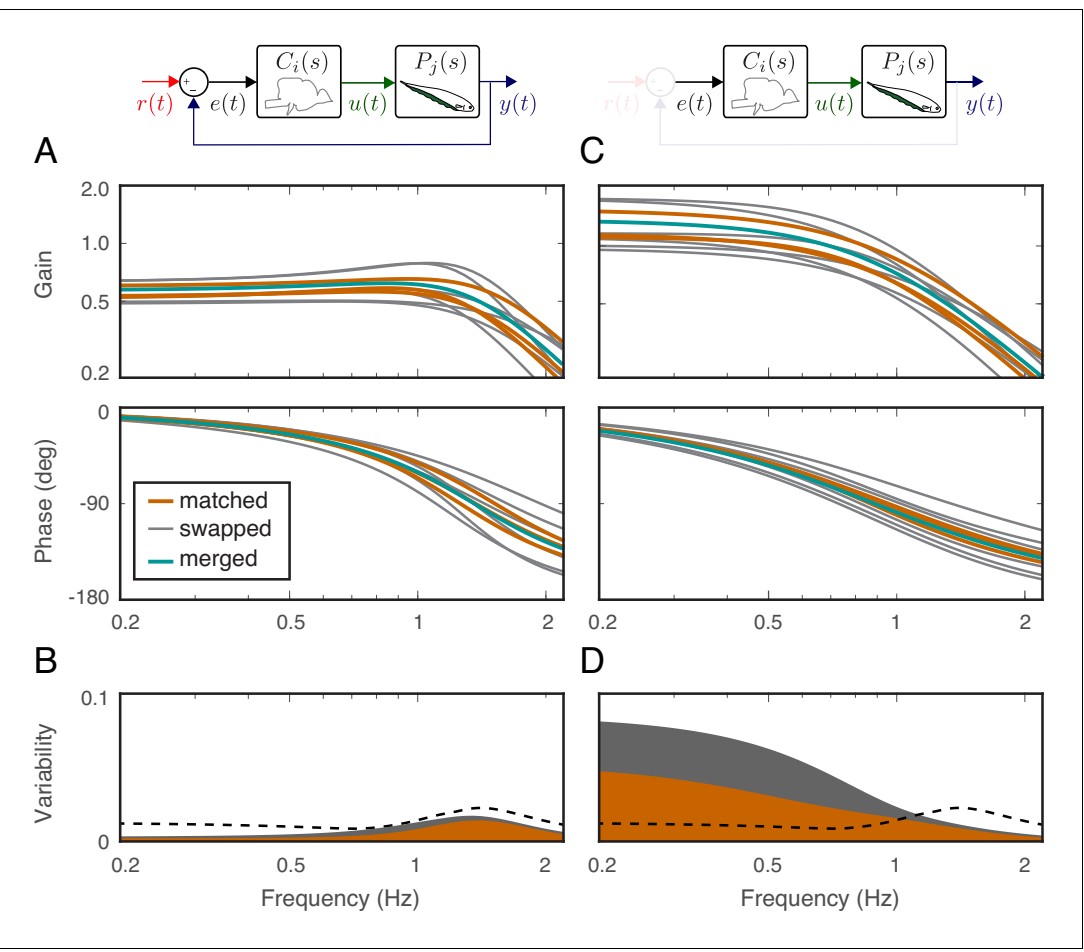

**Figure 4.** Effects of feedback on behavioral variability. At the top of each column is a control diagram that corresponds to all figures in that column. (**A**) Frequency-response gain and phase plots of the estimated input–output models for the matched (orange), swapped (gray), and merged (green) fish in a feedback control topology. (**B**) Variability of the matched (orange) and swapped (gray) models. Dashed line is the behavioral variability observed in the tracking data across fish. (**C**) Same as in (**B**) but for the loop gain, $P(s)C(s)$. (**D**) Same as in (**C**) but for the loop gain, $P(s)C(s)$.

hand, provides a conservative estimate of the variability observed across trials in the real fish. See Model variability and Bootstrap estimate of behavioral variability in Materials and methods.

Using these metrics, we computed the variability of the matched models and compared against the behavioral variability observed in real fish. Unsurprisingly, variability of the matched models (orange region in *Figure 4B*) remained well below the behavioral variability across the entire frequency range of interest (dashed line in *Figure 4B*). Surprisingly, however, the model variability remained below the behavioral variability even for swapped models (*Figure 4B*, gray region versus dashed line). These results highlight the fact that sensory feedback can attenuate the variability of closed-loop models despite mismatch between the controller and plant pairs. In other words, feedback models do not require precise tuning between the controller and plant to achieve the low variability we observed in the behavioral performance of the animals.

Having established that variability of the closed-loop models is robust to the relations between the plant and controller in a feedback system, we examined the role of feedback. This was achieved by examining the loop gain, that is the dynamic amplification of signals that occurs in feedback systems (*Figure 4C*). The only difference between the loop gain model and the closed-loop model is the absence and presence of feedback, respectively. For our model, this was calculated as the product of the plant and controller in both matched and swapped cases. This removal of feedback revealed dramatic variability in the loop gain at frequencies below about 1 Hz (*Figure 4D*). This

variability was well above the behavioral variability observed in fish. In contrast, at frequencies above 1 Hz, the model variability was slightly reduced. These results indicate that sensory feedback attenuates behavioral variability in the biologically relevant range of tracking frequencies at a cost of slightly increased variability at high frequencies.

## Parametrizing the range of neural controllers

As predicted by *Cowan and Fortune (2007)*, each of the feedback controllers obtained above for the averaged fish responses had high-pass filtering characteristics despite the differences in their dynamics (*Figure 5A*). What is the range of neural controllers that, when used in this feedback control topology, leads to behavior that is indistinguishable from the real fish? In other words, how well tuned to the plant does the neural controller need to be to achieve these behavioral performances?

We used the Youla-Kučera parametrization to obtain a range of controllers that generate similar behavioral responses (*Kučera, 2011*). Specifically, this parametrization provided a parametric transfer function describing all stabilizing controllers for the merged plant:

$$C^{\emptyset}(s) = \frac{Q(s)}{1 - P_m(s)Q(s)} \tag{6}$$

Here, $P_m(s)$ is the transfer function of the merged plant and $Q(s)$ is any stable and proper function.

*Equation 6* parametrizes all stabilizing controllers for the merged plant $P_m(s)$. However, we were interested in finding the subset of controllers that yields indistinguishable behavioral performances from real fish. To achieve this, we computed the range for the input–output system dynamics, $G(s)$, of real fish response. Specifically, we calculated the bounds for the gain and phase responses of the 1000 input–output transfer function models estimated while computing the behavioral variability (see Materials and methods). The gray-shaded areas in *Figure 5B* serve as the range of frequency responses that are consistent with behavioral variability of the real fish.

For each of these 1000 transfer functions $G(s)$ that were consistent with the behavioral variability of the fish, we selected $Q(s) = G(s)/P_m(s)$ to generate 1000 corresponding controllers using

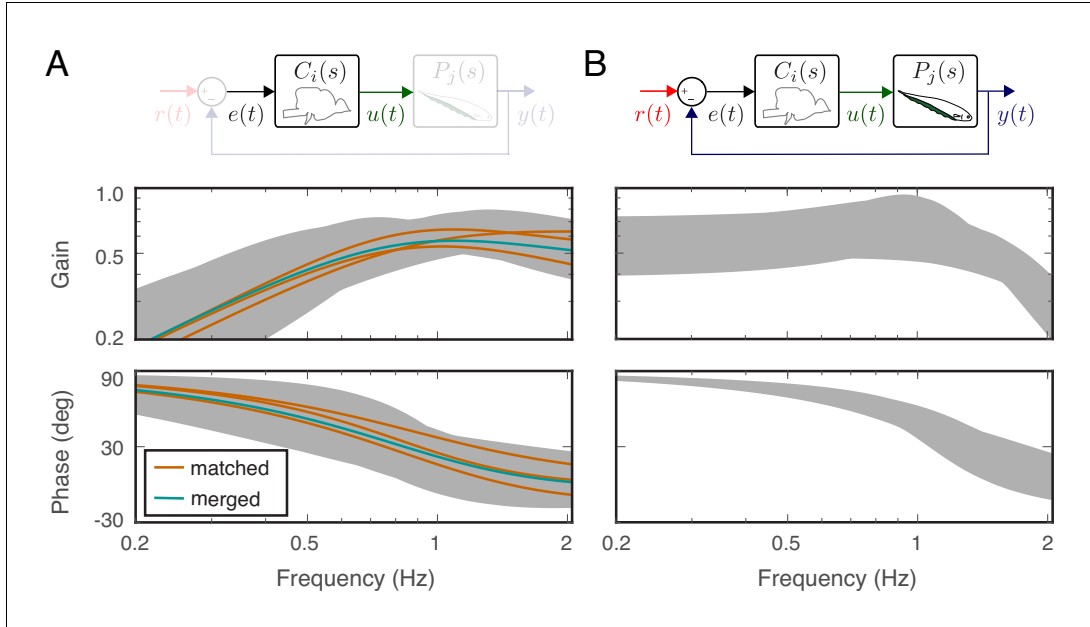

**Figure 5.** Feedback controllers satisfying the behavioral variability. (A) Bode plots for the estimated controllers for the matched (orange) and merged (green) fish under feedback control. Each controller exhibits high-pass filter behavior across the frequency range of interest. Gray shaded regions represent the range of controllers that produce behavioral responses consistent with behavioral variability. (B) The range of behavioral input–output transfer functions consistent with behavioral variability.

*Equation 6*. The bounds for gain and phase responses of these 1000 controllers (the gray shaded regions) show the breadth of controllers that, when implemented within the feedback control topology, produce behavioral outputs consistent with the performance of the real fish (see *Figure 5A*). Note that the controllers calculated in *Equation 3* also satisfy the structure of *Equation 6* when $Q(s) = G(s)/P(s)$ for the associated plant dynamics $P(s)$ of each fish; in other words, the controller estimates are guaranteed to be stabilizing for the associated plants.

These results indicate that the neural controllers need not be precisely tuned to their associated plant dynamics. We found a wide range of controllers that the fish could implement to generate consistent behavioral performance. We note that each of these controllers had high-pass filtering characteristics.

## Discussion

Feedback-based task control allows animals to cope with dramatic but nevertheless common variations of their plant dynamics between individuals. Further, individuals can experience variations in plant dynamics over time such as instantaneous increases in weight during feeding (*Van Handel, 1965*; *Hou et al., 2015*), muscle fatigue during repetitive behaviors (*Enoka and Stuart, 1992*), or carrying heavy objects (*Zollikofer, 1994*). These changes likely result in variable mismatch between the neural controller and the locomotor plant of individual animals. This mismatch is similar to that induced by swapping plants and controllers across individuals, suggesting that moment-to-moment variability can also be eliminated through sensory feedback.

Deciphering the interplay between the task plant, behavioral performance, and neurophysiological activity requires understanding the impacts of the closed-loop control topology. Given the range of morphophysiological features observed across individuals within a species, our results suggest that there is also a range of controller dynamics—ultimately manifest as neurophysiological activity—that each individual could use to achieve consistent biologically relevant behavioral performances. As a consequence, we expect to see more variation at the neurophysiological level than is revealed by task performance for behaviors that rely on closed-loop control.

### Reconciling data-driven and physics-based models of locomotor dynamics

A key contribution of this work is the identification of a data-driven plant model for the locomotor dynamics of a freely behaving animal based on behavioral observations only. To achieve this, we adopted a grey-box system identification approach that seeks to reconcile a physics-based parametric transfer function model with a non-parametric data-driven model (i.e., the frequency-response function).

Developing a model from first principles, for example Newton's laws, is sometimes an effective modeling approach for describing the dynamics of a physical system. For instance, a widely used model in legged locomotion is the spring-loaded inverted pendulum (SLIP) model for describing running dynamics in the sagittal plane (*Blickhan and Full, 1993*; *Full and Koditschek, 1999*). While physics-based models have proven to be successful in modeling the dynamics of biological and mechanical movements, there are limitations. Physics-based approaches for modeling behaviors at lower levels (e.g., the spiking activity of all motor neurons) may lead to a very complex model that does not accurately capture high-level behavior.

Data-driven system identification approaches are used to directly identify a dynamical model based on empirical data (*Ljung, 1998*; *Kiemel et al., 2011*; *Uyanik et al., 2019a*). In general, data-driven system identification may take a black-box approach in which only a general model structure is assumed (say, an ODE or frequency response function). However, data-driven techniques typically generate numerical transfer function estimates to represent animal behavior.

Alternatively, the so-called grey-box approach that we adopt in this paper integrates the structure of a specific physics-based model but leaves its parameters free, relying on data-driven system identification to fit those parameters. In this case, prior knowledge about the underlying dynamical model informs and constrains data-driven system identification. Grey-box identification can provide a bridge between top-down, data-driven modeling and bottom-up, physics-based modeling. We utilized the parametric dynamical model of *Sefati et al. (2013)* for the plant but estimated the model

parameters using data-driven system identification techniques. Our results show that the data-driven estimates for the plant dynamics match the structure of this model (*Figure 2A*).

## Effects of variability in plant dynamics

Our results reveal two complementary perspectives on variability in plant dynamics. On the one hand, estimates of the closed-loop controllers were highly sensitive to the dynamics of the plant of individual fish. This was an inevitable consequence of our strategy for estimating the controllers—inferring the controllers from the plant and closed-loop dynamics. On the other hand, our closed-loop control models were robust to variability of either the plant or controller, indicating that precise tuning is not needed for this behavior. A control-theoretic sensitivity analysis demonstrates that these results are not unique to this example but rather are a general property of feedback control systems, see *Csete and Doyle (2002)* for a review.

Specifically, consider the frequency dependent sensitivity function of the feedback controller $C(s)$ with respect to plant $P(s)$, in the closed-loop topology:

$$S_{(C \leftarrow P)}(j\omega) = \frac{\partial C}{\partial P} = \left[ \frac{G(j\omega)}{G(j\omega) - 1} \right] \frac{1}{P^2(j\omega)} \tag{7}$$

The sensitivity of the controller to the plant dynamics is a frequency dependent function. It depends on the gain and phase of both the measured closed-loop transfer function $G(j\omega)$ as well as the plant model $P(j\omega)$. At low frequencies, fish track extremely well and thus $G(j\omega) - 1 \approx 0$. At high frequencies, the low-pass plant $P(j\omega)$ is small. Combining these factors, we expect the sensitivity $|S(j\omega)|$ to be large across frequencies. In other words, there is an inescapable sensitivity to plant dynamics when the controllers are estimated using this computational strategy.

We conducted a complementary analysis to compute the sensitivity of the closed-loop tracking response $G$ to perturbations in the combined controller and plant dynamics $PC$. We treated the controller–plant pair $PC$ as a single variable and obtained the frequency-dependent sensitivity function as

$$S_{(G \leftarrow PC)}(j\omega) = \frac{\partial G}{\partial PC} = \frac{1}{(1 + PC(j\omega))^2}. \tag{8}$$

At low frequencies, $G(j\omega)$ has nearly unity gain and thus $PC(j\omega)$ goes to $\infty$. As a result, sensitivity $S_{(G \leftarrow PC)}$ approaches zero. At high frequencies, $PC(j\omega)$ goes to zero and thus sensitivity is bounded around 1. Thus, despite the fact that the controller estimates are sensitive to plant variations, the closed-loop transfer function (in the presence of sensory feedback) is robust against variability of controller–plant pairs.

## Feedback and variability in neural systems

These findings suggest that a fish could implement a range of controllers in its nervous system for refuge tracking. These controllers must have high-pass filtering characteristics, but their details may be inconsequential. This has two implications for neurophysiological analysis of neural control systems. First, neurophysiological activity within control circuits in open-loop experiments (e.g. playback and anesthetized/immobilized animals) need not appear to be well tuned for the control of a behavioral performance. This poor tuning, which may manifest in variability that appears across levels of functional organization—from variability in neural activity within neurons, variability in tuning across neurons, and variability across individuals—is refined via feedback during behaviors in which the feedback loops are intact. Second, there must be mechanisms by which the controllers are slowly (at time constants greater than that necessary for the moment-to-moment control of the behavior) tuned to the dynamics of the animal's locomotor plant. For instance, adaptation of cerebellar activity in relation to mismatch between intended versus actual motor performances contribute to the retuning of neural controllers (*Morton and Bastian, 2006*; *Bell et al., 1997*; *Pisotta and Molinari, 2014*).

Feedback is mediated both through the effects of behavior on sensory receptors and via descending pathways in the brain. Behavior generally results in concomitant activation of receptor types across the animal, which can include, for example, simultaneous stimulation of stretch receptors embedded in muscles and visual receptors in eyes. Correlations in feedback-related activity across sensory modalities likely contribute to robust control (*Roth et al., 2016*). Internal feedback

pathways, interestingly, have been recently shown to synthesize sensory filtering properties of behaviorally-relevant stimuli. Descending neural feedback is used to dynamically synthesize responses to movement (*Metzen et al., 2018*; *Huang et al., 2018*; *Clarke and Maler, 2017*).

How do other animal systems manage variability, broadly speaking, to achieve consistent output? In pyloric neural circuits of crustaceans, the oscillatory output of the system is consistent despite dramatic variations in the dynamics of cellular and membrane properties of neurons within these circuits (*Goaillard et al., 2009*; *Marder and Taylor, 2011*). How these circuits maintain consistent output despite underlying variability remains an open question (*Hamood and Marder, 2014*) but likely relies on feedback regulation that is intrinsic to the neural network itself.

The mechanisms by which systems maintain consistent output can be assessed through behavioral analysis (*Krakauer et al., 2017*) of responses to systematic perturbations (*Cowan et al., 2014*). For example, perturbations have been used in many species including flying insects (*Bender and Dickinson, 2006*; *Matthews and Sponberg, 2018*), walking sticks (*Dallmann et al., 2019*; *Diederich et al., 2002*), and humans (*Lee and Perreault, 2019*) to reveal how control systems manage mechanical and sensory variation. These analyses and others show that animals rely on sensory feedback to modulate moment-to-moment movement to maintain consistent task performances.

Ultimately, understanding how robustness emerges in closed loop requires investigating the interplay between plants and controllers, which are inextricably linked (*Cowan and Fortune, 2007*; *Tytell et al., 2011*; *Tytell et al., 2018*). A commonly implemented strategy, for example, is the use of low-pass plant dynamics. This strategy can avoid instabilities that arise from long-latency feedback, an inescapable feature of biological control systems (*Sponberg et al., 2015*; *Madhav and Cowan, 2020*). Specifically, delay introduces phase lag that increases with frequency. As the phase angle of the loop gain $PC$ approaches $-180°$, the likelihood of instability increases. A low-pass plant can mitigate this instability by ensuring that the gain of $|PC| \ll 1$ near this 'phase crossover frequency.' In turn, the animal exhibits low behavioral sensitivity, namely $S_{(G \leftarrow PC)} \leq 1$ in *Equation 8*. In short, a well-tuned neuromechanical plant can simplify feedback control by rendering the dynamics passively self-stabilizing (*Hedrick et al., 2009*; *Sefati et al., 2013*), while nevertheless maintaining behavioral flexibility (*Cowan et al., 2014*).

## Materials and methods

All experimental procedures with fish were reviewed and approved by the Johns Hopkins University and Rutgers University Animal Care and Use Committees and followed guidelines for the ethical use of animals in research established by the *National Research Council* and the *Society for Neuroscience*.

Adult *Eigenmannia virescens*, a species of weakly electric Gymnotiform fish, were obtained through commercial vendors and housed in laboratory tanks. Tanks were maintained at a temperature of approximately 27°C and a conductivity between $50 - 200$ μS . We transferred individual fish to the experimental tank about 1 day before experiments for acclimation. Three fish were used in this study.

### Experimental apparatus

The experimental apparatus is similar to that used in previous studies (*Stamper et al., 2012*; *Biswas et al., 2018*; *Uyanik et al., 2019b*). A refuge machined from a PVC pipe with a length of 15 cm and 5.08 cm diameter was placed in the experimental tank with the fish. The bottom face of the refuge was removed to allow video recording from below. Six windows, 0.625 cm in width and spaced within 2.5 cm intervals, were machined onto each side to provide visual and electrosensory cues. During experiments, we actuated the refuge using a linear stepper motor with 0.94 μm resolution (IntelLiDrives, Inc Philadelphia, PA, USA) driven via a Stepper motor controller (Copley Controls, Canton, MA, USA). MATLAB (MathWorks, Natick, MA, USA) scripts were used to control the movement of the refuge and to capture video. Video data were captured using a pco.1200s high speed camera (Cooke Corp., Romulus, MI, USA) with a Micro-Nikkor 60 mm f/2.8D lens (Nikon Inc, Melville, NY, USA). All videos used for data analysis were shot at 30 frames per second with $1280 \times 1024$ pixel resolution. Some videos of ribbon fin motion were shot at 100 frames per second.

## Experimental procedure

Refuge movement consisted of single sine waves of amplitude 0.1 cm and of frequencies 0.55, 0.95, and 2.05 Hz. The amplitude of refuge movements was chosen because fish rely on counter propagating waves for tracking in this regime (*Roth et al., 2011*). At higher amplitudes, fish often will use a uni-directional wave in the ribbon fin for locomotion. The frequencies were selected to be within the normal tracking regime as determined in previous studies (*Stamper et al., 2012*; *Biswas et al., 2018*; *Uyanik et al., 2019b*). Trials were randomized with respect to frequency. Each trial lasted 60 seconds. The stimulus amplitude was linearly ramped up over the first ten seconds to prevent startle responses from the fish. During the experimental phase, the stimulus frequency and amplitude were maintained for 40 seconds. Finally, the stimulus amplitude was ramped down during the final ten seconds. Trials were separated by a minimum break of 2 minutes.

## Derivation of plant model

*Sefati et al. (2013)* developed and tested a second-order, linear, ordinary differential equation that describes how changes in fore-aft position of the animal, $y(t)$, relate to changes in position of the nodal point, $u(t)$:

$$m\frac{d^2y}{dt^2} + b\frac{dy}{dt} = ku(t) \tag{9}$$

Here, $m$, $k$, and $b$ represent mass, gain, and damping, respectively. This equation follows from Equation S13 in the Supporting Information of *Sefati et al. (2013)*. Note that the present paper uses slightly different nomenclature; in particular $u(t)$, $y(t)$, and $b$ in the present paper correspond to $\Delta L$, $x(t)$, and $\beta$, respectively, in *Sefati et al. (2013)*.

The Laplace transform provides a computationally convenient means to represent dynamics of linear, time-invariant systems such as the one in *Equation 9* (*Roth et al., 2014*). Taking the Laplace transform of both sides of *Equation 9*, neglecting initial conditions, and algebraically simplifying, we arrive at the plant model in *Equation 1*:

$$\frac{Y(s)}{U(s)} = \overbrace{\frac{k}{ms^2 + bs}}^{P(s)} \tag{10}$$

## Inferring controller using plant and closed-loop dynamics

Given the feedback control topology in *Figure 1*, the closed-loop dynamics relating the movement of the refuge to the movement of the fish are given in the Laplace domain by the following equation:

$$G(s) = \frac{P(s)C(s)}{1 + P(s)C(s)} \tag{11}$$

This equation is also shown in Equation 7 of *Cowan and Fortune (2007)* with a slightly different nomenclature; in particular $P(s)$ and $G(s)$ in the present paper correspond to $G(s)$ and $H(s)$, respectively, in *Cowan and Fortune (2007)*.

Given $G(s)$ and $P(s)$, one can compute the complementary controller $C(s)$ using *Equation 11* as

$$C(s) = \frac{G(s)}{(1 - G(s))P(s)}. \tag{12}$$

## Reconciling data-driven and physics-based approaches to estimate the locomotor dynamics

The position of the refuge and fish were tracked for each video using custom software (*Hedrick, 2008*). The videos were analyzed to extract 3 to 10 seconds segments, where the fish used counter propagating waves for refuge tracking. Then, the nodal point was hand clicked in these video segments: 18,000 nodal point measurements were made over a total of 106 segments of data.

The physics-based plant model in *Sefati et al. (2013)* was previously validated with quasi-static open-loop experiments. Here we reconciled the physics-based plant model from *Sefati et al. (2013)* (*Equation 1*) with the data that were collected in tracking experiments.

For each frequency of refuge movement, M segments of nodal point data were extracted. Each segment of data consists of the following measurements: nodal point shift $\{u_1^m, u_2^m, \ldots, u_n^m\}$ and fish position $\{y_1^m, y_2^m, \ldots, y_n^m\}$ as a function of time $\{t_1^m, t_2^m, \ldots, t_n^m\}$, where n is the number of samples and $m = \{1, 2, \ldots, M\}$.

We estimated the magnitude and phase of the plant model for each frequency of refuge movement. The average value of nodal point shift and fish position were computed from M data segments per fish for each frequency of refuge movement. We aligned each data segment based on the phase of refuge signals. The segments are not completely overlapping: we selected the largest time window with at least 50 percent overlap of data segments. A sine wave function of the following form was fitted to the average nodal point data, $u_{avg}(t)$, and average fish position data, $y_{avg}(t)$, as

$$u_{avg}(t) = A_u \sin(2\pi f_i t + \phi_u) + B_u, \tag{13}$$

$$y_{avg}(t) = A_y \sin(2\pi f_i t + \phi_y) + B_y, \tag{14}$$

where input–output pairs $(A_u, A_y)$, $(\phi_u, \phi_y)$ and $(B_u, B_y)$ correspond to magnitudes, phases and DC offsets in polar coordinates, respectively. Note that this fitting was done separately for each refuge frequency, $f_i = \{0.55, 0.95, 2.05\}$ Hz.

After computing the magnitude and phase for both the average nodal shift and fish position, we estimated the magnitude and phase for the plant transfer function at $\omega_i = 2\pi f_i$ as

$$|\hat{P}(j\omega_i)| = \frac{A_y}{A_u}, \tag{15}$$

$$\angle \hat{P}(j\omega_i) = \phi_y - \phi_u, \tag{16}$$

We obtained a non-parametric estimate of the plant transfer function for each frequency $\omega_i$, that is $\hat{P}(j\omega_i)$ by estimating magnitude and phases. We used $\hat{P}(j\omega_i)$ to estimate the parameters of the transfer function model given in *Equation 1*. In this model, there are three unknown parameters, namely $m$, $k$, and $b$. However, for the fitting purposes we reduced the number of unknown parameters to two by normalizing the 'gain' ($k$) and 'damping' ($b$) by the 'mass' ($m$). The normalized plant transfer function in Fourier domain takes the form

$$P(j\omega) = \frac{k/m}{(j\omega)^2 + (b/m)(j\omega)} = \frac{k/m}{-\omega^2 + (bj\omega/m)}. \tag{17}$$

where $j = \sqrt{-1}$. For an ideal deterministic system, for each frequency $P(j\omega_i) = \hat{P}(j\omega_i)$, where $\hat{P}(j\omega_i)$ corresponds to the non-parametrically computed frequency response function. For this reason, estimates of transfer function parameters were made by minimizing a cost function using gradient descent method:

$$J(k/m, b/m) = \sum_{i=1}^{3} |P(j\omega_i) - \hat{P}(j\omega_i)|^2. \tag{18}$$

## Bootstrap estimate of behavioral variability

We estimated behavioral variability using bootstrap estimates derived from individual experimental trials at the three test frequencies. Across all three fish, we made 37 observations of the frequency response at $f_i = 0.55$ Hz, 35 observations at $f_i = 0.95$ Hz, and 34 observations at $f_i = 2.05$ Hz, namely:

$$\begin{aligned}
&\{\hat{G}_1(j2\pi 0.55), \hat{G}_2(j2\pi 0.55), \ldots, \hat{G}_{37}(j2\pi 0.55)\} \\
&\{\hat{G}_1(j2\pi 0.95), \hat{G}_2(j2\pi 0.95), \ldots, \hat{G}_{35}(j2\pi 0.95)\} \\
&\{\hat{G}_1(j2\pi 2.05), \hat{G}_2(j2\pi 2.05), \ldots, \hat{G}_{34}(j2\pi 2.05)\}
\end{aligned} \tag{19}$$

To estimate the behavioral variability at frequencies that were not explicitly tested, we used a parametric approach. Specifically, we constructed $N = 1000$ triplets by randomly selecting one frequency response function from each of the test frequencies in *Equation 19*. For each of the 1000

triplets, we estimated a transfer function, $G_{\text{bootstrap},i}(s)$, $i = 1, \ldots, N$ of the form in **Equation 2** using Matlab's transfer function estimation method 'tfest'.

Let $x_i$ and $y_i$ be real and imaginary parts of the complex-valued frequency response function, namely $G_{\text{bootstrap},i}(j\omega_0) = x_i + jy_i$, where $\omega_0$ is a frequency in the range 0.2 Hz to 2.05 Hz. The covariance matrix for the estimated frequency response function in the complex domain was calculated as

$$\text{Cov}^{\omega_0} = \begin{bmatrix} \sigma_{xx}^{\omega_0} & \sigma_{xy}^{\omega_0} \\ \sigma_{yx}^{\omega_0} & \sigma_{yy}^{\omega_0}, \end{bmatrix} \tag{20}$$

where

$$\sigma_{xx}^{\omega_0} = \frac{1}{N-1} \sum_{i=1}^{N} (x_i - \mu_x)^2, \tag{21}$$

$$\sigma_{yy}^{\omega_0} = \frac{1}{N-1} \sum_{i=1}^{N} (y_i - \mu_y)^2, \tag{22}$$

$$\sigma_{xy}^{\omega_0} = \sigma_{yx}^{\omega_0} = \frac{1}{N-1} \sum_{i=1}^{N} (x_i - \mu_x)(y_i - \mu_y). \tag{23}$$

Here, $\mu_x$ and $\mu_y$ are mean values of $x_i$ and $y_i$, $\forall i \in \{1, 2, \ldots, N\}$, respectively. The final bootstrap estimate of behavioral variability was calculated as the largest singular value of the central covariance matrix, $\text{Cov}^{\omega_0}$. In addition, the range of the gain and phase of these 1000 transfer function models was plotted in **Figure 5B**.

## Model variability

The variability across 'matched' and 'swapped' models was calculated for both the closed-loop transfer function $G(s)$ (**Figure 4B**) and the loop gain $P(s)C(s)$ (**Figure 4D**). We evaluated each of the three fish-specific controllers and plant transfer functions at frequencies between 0.2 Hz and 2.05 Hz; for each frequency $\omega_0$ in this range, we have $C_1(j\omega_0)$, $C_2(j\omega_0)$, and $C_3(j\omega_0)$ and $P_1(j\omega_0)$, $P_2(j\omega_0)$, and $P_3(j\omega_0)$.

To calculate the matched closed-loop variability, we first calculated the real ($x_i$) and imaginary ($y_i$) parts of $C_i(j\omega_0)P_i(j\omega_0)/(1 + C_i(j\omega_0)P_i(j\omega_0))$. Using these values, the matched closed-loop variability was calculated as the largest singular value of the central covariance matrix of these ordered pairs. The matched loop-gain variability was calculated similarly, using the real and imaginary parts of $C_i(j\omega_0)P_i(j\omega_0)$. For each of these calculations, $N = 3$, because there are three sets of matched models. The closed-loop and loop-gain swapped variability was calculated identically, except using the $N = 6$ swapped permutations of control–plant pairs. **Figure 4B,D** illustrates the variability of the matched and swapped models both for closed-loop control and loop gain.

## Acknowledgements

This work was supported by a collaborative National Science Foundation (NSF) Award to Noah J Cowan (1557895) and Eric S Fortune (1557858). Figures and Figure Supplements incorporate illustrations drawn by Eatai Roth. We also thank him for his invaluable feedback on the manuscript.

## Additional information

### Funding

| Funder | Grant reference number | Author |
| --- | --- | --- |
| National Science Foundation | 1557895 | Noah J Cowan |
| National Science Foundation | 1557858 | Eric S Fortune |

The funders had no role in study design, data collection and interpretation, or the decision to submit the work for publication.

### Author contributions
Ismail Uyanik, Conceptualization, Data curation, Software, Formal analysis, Validation, Investigation, Visualization, Methodology; Shahin Sefati, Conceptualization, Data curation, Software, Formal analysis, Validation, Visualization, Methodology; Sarah A Stamper, Conceptualization, Data curation, Formal analysis; Kyoung-A Cho, Data curation; M Mert Ankarali, Software, Formal analysis; Eric S Fortune, Noah J Cowan, Conceptualization, Resources, Supervision, Funding acquisition, Validation, Investigation, Visualization, Methodology, Project administration

### Author ORCIDs
Ismail Uyanik (iD) https://orcid.org/0000-0002-3535-5616
Eric S Fortune (iD) https://orcid.org/0000-0001-6447-5425
Noah J Cowan (iD) https://orcid.org/0000-0003-2502-3770

### Ethics
Animal experimentation: All experimental procedures used for this study were reviewed and approved by Johns Hopkins (protocol: FI19A178) and Rutgers (protocol: 999900774) Animal Care and Use committees and followed the guidelines given by the National Research Council and the Society for Neuroscience.

### Decision letter and Author response
Decision letter https://doi.org/10.7554/eLife.51219.sa1
Author response https://doi.org/10.7554/eLife.51219.sa2

## Additional files

### Supplementary files
• Transparent reporting form

### Data availability
An archived version of the dataset and analysis code is available through the Johns Hopkins University Data Archive.

The following dataset was generated:

| Author(s) | Year | Dataset title | Dataset URL | Database and Identifier |
|---|---|---|---|---|
| Uyanik I, Sefati S, Stamper SA, Cho K-A, Ankarali MM, Fortune Eric S, Cowan NJ | 2020 | Data associated with Variability in locomotor dynamics reveals the critical role of feedback in task control | https://doi.org/10.7281/T1/UDTJPD | Johns Hopkins University Data Archive, 10.7281/T1/UDTJPD |

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
