## [Decision Letter]

**Acceptance summary:**

The authors ask whether the neural controller (used when an animal engages in precise behavior) is optimized for each animal’s particular motor plant properties or whether a generic controller is sufficient and instead robust feedback mechanisms ensure that the generic controller works well despite variations in the motor plant. The authors have found a very clever way to address this problem in the refuge tracking behavior of an electric fish. The CNS controller controls the location of a 'nodal point' along the ribbon fin of these fish. Forward- and backward-directed oscillations of the ribbon fin (opposing forces) meet at the nodal point and the fish can precisely control the location of the nodal point in order to achieve both stability and maneuverability during locomotion. The movement of the nodal point serves as a linear estimate of the controller output of the fish as it tracks the moving refuge. Kinematic measurements were used to generate individualized estimates of each fish's locomotor plant and controller revealing substantial variability between fish. To test the impact of this variability on behavioral performance, these models were used to perform simulated 'brain transplants' – computationally swapping controllers and plants between individuals. The authors conclude that simulated closed-loop performance was robust to mismatch between plant and controller. This seminal observation suggests that animals rely on feedback rather than precisely tuned neural controllers to compensate for morphophysiological variability.

**Decision letter after peer review:**

Thank you for submitting your article "Variability in locomotor dynamics reveals the critical role of feedback in task control" for consideration by *eLife*. Your article has been reviewed by three peer reviewers, and the evaluation has been overseen by Ronald Calabrese as the Senior and Reviewing Editor. The following individual involved in review of your submission has agreed to reveal their identity: Len Maler (Reviewer #1).

The reviewers have discussed the reviews with one another and the Reviewing Editor has drafted this decision to help you prepare a revised submission.

Summary:

This is an impressive paper and builds on a series of impressive papers. The quality of the experiments and analyses is very high. The authors' perspective is from the engineering approach in which the motor activity of, e.g., a robot or weakly electric fish is divided into the mechanics (motor plant) and 'controller' (sensory system and CNS). Furthermore, the mechanics will vary across different animals due to their size etc. The question posed by the authors is whether the controller is optimized for each animal for its particular motor plant properties vs. whether a generic controller is sufficient and robust feedback mechanisms ensure that the generic controller works well despite variations in the motor plant. This is a very general question and one that is hard or impossible to address in most animal models. As carefully argued by the authors, devising separate data-driven models of controller and plant is very difficult for muscle-based plants simply because most motor actions (e.g., stepping) require coordination of many muscles across joints.

The authors, based on prior papers, have found a very clever way to circumvent this problem in the refuge tracking behavior of a weakly electric fish (*Eigenmannia*). Their earlier studies showed that the CNS controller operated by controlling the location of a 'nodal point' along the ribbon fin of these fish. Forward- and backward-directed oscillations of the ribbon fin (opposing forces) meet at the nodal point and the fish can precisely control the location of the nodal point in order to achieve both stability and maneuverability during locomotion. The position of the nodal point varies as a function of swim speed. The authors argue that the movement of the nodal point can serve as a linear estimate of the controller output of the fish as it tracks the moving refuge. Kinematic measurements of *Eigenmannia* were used to generate individualized estimates of each fish's locomotor plant and controller revealing substantial variability between fish. To test the impact of this variability on behavioral performance, these models were used to perform simulated 'brain transplants' – computationally swapping controllers and plants between individuals. The authors conclude that simulated closed-loop performance was robust to mismatch between plant and controller. This suggests that animals rely on feedback rather than precisely tuned neural controllers to compensate for morphophysiological variability.

Essential revisions:

1) There is a consensus that the manuscript is too dense with many points needing to be clarified as detailed in the attached reviews (Reviewer #2 in particular). We do think that this paper should be in a general journal like *eLife*, partially because it is important that neurobiologists become at least familiar with the concepts and, to a limited extent, to the methods of control theory as practiced by engineers. Right now, the manuscript just won't work for most neurobiologists. But it can be written in a much clearer way. The import of the equations should be explained in the text and the derivations placed in a supplementary information section for specialists.

2) Might the model be tested directly/experimentally by perturbing the motor plant, for example weighting the fish by overfeeding or with a weight belt (Reviewer #3)? There is a realization that these approaches might be unsuitable, but the experimenters are resourceful, and some perturbation might be found, e.g. clipping some of the ribbon fin (caudal quarter maybe).

Reviewer #1:

This is an impressive manuscript and built on a series of impressive papers. The quality of the experiments and analyses are very high and I have no serious concerns about the conclusions drawn with one caveat (see below).

The authors’ perspective is from the engineering approach in which the motor activity of, e.g., a robot or weakly electric fish is divided into the mechanics (motor plant) and 'controller' (sensory system and CNS). Furthermore, the mechanics will vary across different animals due to their size etc. The question posed by the authors is whether the controller is optimized for each animal with respect to its particular motor plant properties vs. whether a generic controller is sufficient and robust feedback mechanisms ensure that the generic controller works well despite variations in the motor plant.

This is a very general question and also one that is hard or impossible to address in most animal models. As carefully argued by the authors, devising separate data-driven models of controller and plant is very difficult for muscle-based plants simply because most motor actions (e.g., stepping) require coordination of many muscles across joints.

The authors, based on prior papers, have found a very clever way to circumvent this problem in the refuge tracking behaviour of a weakly electric fish (*Eigenmannia*). Their earlier studies showed that the CNS controller operated by controlling the location of a 'nodal point' along the ribbon fin of these fish. Forward- and backward-directed oscillations of the ribbon fin (opposing forces) meet at the nodal point and the fish can precisely control the location of the nodal point in order to achieve both stability and maneuverability during locomotion. The position of the nodal point varies as a function of swim speed. The authors argue, and I agree, that the movement of the nodal point can serve as a linear estimate of the controller output of the fish as it tracks the moving refuge.

Given this assumption and the authors’ prior papers, the experiments were well justified and well carried out. The analysis is, I expect, also well carried out. But here I think that the authors simply skipped too many steps in their derivations. This is the major problem that I think must be addressed in a revision.

1) "We adopted the physics-based parametric model of locomotor dynamics of *Eigenmannia* described by Sefati et al., 2013", this is the 'controller dynamics'. This is Equation 1 in the Results and gives P(s). I think this is derived from Equation 4 is Sefati et al., 2013 (derived very nicely in the supplementary information). But, I fail to see the derivation. It is true that I did not expend a lot of effort trying to derive Equation 3 from Equation 4; however, I think that a typical reader should not have to make this attempt. The authors should make the derivation explicitly, perhaps in a supplementary information section (as was done in Sefati et al., 2013).

2) The authors then use a previous paper (Cowan and Fortune, 2007) to represent the behavioural response of the fish (Equation 2); this is well justified and gives G(s). The next point is the most critical and most novel. The authors then 'back calculate' the parameters for the controller, P(s), by using data driven behavioural model encapsulated in G(s). This results in Equation 3. At this point, Cowan and Fortune is cited and I got confused. Their closed loop transfer function, H(s), has an entirely different form from Equation 3. I am presumably missing something here and I think the authors really have to spell out in detail how the do the 'back calculation' and how it connects to Cowan and Fortune's results. Again, this would likely be a fairly detailed derivation and, as in the Sefati et al. paper, could be presented in supplementary information.

3) Lastly, I was struck by the parallel between the authors' conclusions and that of Marder and colleagues. Dr. Marder and colleagues have shown, over many papers, that the output of the STG pattern generator can be produced by many different (with respect to parameters) STG models. There is no unique STG pattern generator and the average parameters don't match any real STG. A paper by O'Leary and Marder (2016, Current Biology) advances a similar argument to the one in this manuscript – that "The high variability in conductance densities in these neurons [9, 10] appears to contradict the possibility that robustness is achieved through precise tuning of key temperature-dependent processes."

This paper uses Ca-dependent global feedback (Ca entry depends on activity) to maintain robust STG activity over a wide range of temperatures, despite the individual ion channels not being tuned for this temperature range. Although the systems are entirely different the principle seems, to me, to be the same. It would be great to add this to the Discussion.

One overall conclusion of this paper is that "Physics-based approaches for modeling behaviors at lower levels (e.g., the spiking activity of all motor neurons) may lead to a very complex model that does not accurately capture high-level behavior." I fully agree and I also am convinced that the feedback tuning mechanisms of this manuscript and the O'Leary and Marder paper are the way to go for modelling complex systems – when one is dealing with even the electrosensory system of *Eigenmannia*, there are too many parameters to be estimated for a realistic approach to neural models. This is a powerful conclusion and giving it the wider context – crab STG and *Eigenmannia* electrosensory system and ribbon fin.

Reviewer #2:

In this paper the authors build on their prior work examining the neuromechanical control of locomotion in *Eigenmannia* to attack an important and understudied question in motor control – how neural control strategies vary based on individual differences in the motor plant across individuals. To do so, the authors combine detailed behavioral measurements with control-theoretic analysis and simulations to argue that animals use feedback (sensory) signals, rather than precisely-tuned feedforward neural controllers, to robustly control behavior.

This is a potentially highly impactful paper, however, I have three major concerns as listed below. Also, I should provide the caveat that I am neither an expert in models of fish locomotion nor in the kind of control-theoretic analyses used here. Therefore I must leave the technical evaluation of these aspects to other reviewers.

1) Comprehensibility of this paper to a general audience of biologists. This is a highly technical paper which the authors are trying to squeeze into 4 main-text figures and the absolute minimum of explanatory text. Because of this choice, I think that this paper will not be comprehensible to most *eLife* readers (in its present state this seems much more appropriate for PLOS Comp Bio or some more specialized computational journal), thereby reducing its potential impact. I think the authors need to expand the text (and possibly include a couple of explanatory figures) to allow readers without control systems expertise to understand the paper. For example, a reader not already familiar with Nyquist or Bode plots will not be able to understand any of the key evidence for the authors' arguments presented in Figures 3 or 4. In the main text, the authors often cite their prior work rather than explaining essential components of their approach. This includes small things like not defining/explaining the variables in Equation 2 as well as explaining the key strengths of their model system – it took me several readings to understand what was meant by the nodal point providing a "linear proxy" for u(t).

2) Dissociating the role of feedback control from that of plant dynamics. The authors present the study as attempting to distinguish between two alternatives – that behavioral robustness arises from either precise feedforward neural control or well-tuned feedback control. However it's not clear to me whether or how the authors have ruled out a third possibility – that the physics of the plant are such that behavioral output (forward-backward movement in the refuge task) is simply insensitive to variations in the motor command, at least over the range of motor command variations actually produced by the fish. In fact the result shown in Figure 3C would appear to be consistent with this possibility. The authors offer this interpretation:

"Surprisingly, however, the variability remained below the trial-to-trial variability (Figure 3C, gray region versus dashed line). These results highlight the fact that sensory feedback can attenuate the output variability despite mismatch between the controller and plant pairs. In other words, feedback models do not require precise tuning between the controller and plant to achieve the low variability we observed in the behavioral performance of the animals."

It seems to me that an equally plausible explanation is not that sensory feedback is compensating for "mismatch" errors, but rather that the physical plant is simply insensitive to errors. How can the authors rule this out? An analysis quantifying how much the natural variations in u(t) actually cause variations in y(t) would seem critical to answer this. It's possible that this issue is already implicitly addressed in the analyses provided, but if so I can't figure this out and I'd like the authors to clarify.

3) Expanding on a couple of Discussion points

The authors conclude the Discussion as follows:

"The dynamic synthesis of filters based on current sensory information may be a mechanism that shapes neural controllers on two time scales: for the moment-to-moment task dynamics and over the longer term for the maintenance of behavioral performances."

I'd suggest that the authors expand on this idea a bit; at present it's not very clear to me how they think their simulation results (e.g. that the synthesis of the high-pass controller and low-pass plant yields a robust system) relate to the question of multiple timescales.

Also I'd encourage the authors to expand on possible biological implementations of their findings, and speculate a bit on how those implementations might be similar or different from those known in other model systems. For instance, the controller described here is able to produce similar results (i.e., tracking the same stimulus albeit with greater variability) with different parameters (i.e., switched case) and is reminiscent of multifunctional neural circuits such as those characterized in the crustacean pyloric system. In addition, the characterization of how feedback reduces variability is similar to how the role of feedback and CPG entrainment is studied in multiple species including cats, stick insects, and crayfish. To be clear I'm not suggesting that the authors review all of these (or even any of them in great depth), but I'm sure readers will wonder how the authors’ theories relate to ideas from other systems.

Reviewer #3:

I've been thinking a lot about this paper and while I argue that it should eventually be published in *eLife*, I think (for this broad readership journal) it could use some expansion regarding pedagogy on the control aspects. I find it very challenging to decide if I believe that the "simulated swapping" tells me anything fundamental about the system, or if it is just a computational exercise. I think it is not, but it is hard for me to evaluate given all the assumptions going into "gray box" model. What if the black box model isn’t the correct one? In particular, since the authors likely have room in terms of word count, and certainly have space in terms of numbers of figures, I think a bit more explanation of the methods to demystify the story would be helpful.

Based on the first paragraph of the Introduction, I would have expected to see how the controller changes when fish lose or gain weight (or some manipulation). But I am not convinced by their statement "These changes likely result in variable mismatch between the neural controller and the locomotor plant of individual animals. This mismatch is similar to that induced by swapping plants and controllers across individuals, suggesting that moment-to-moment variability can also be eliminated through sensory feedback." Is this true? Why not do the (obviously harder) experiment of indeed letting the fish eat a lot (or not) and measuring the controller? That way, the authors don't have to do a virtual swap. Even better, why not add weighted "jackets" to the fish as a real "plant" manipulation? I think if they could show this, the paper would become outstanding (instead of "just" very cool as it is now).

This phrase in the Abstract "virtually indistinguishable between conspecifics" is a bit overwrought. What does "virtually indistinguishable" mean?

---

## [Author Response]

Essential revisions:1) There is a consensus that the manuscript is too dense with many points needing to be clarified as detailed in the attached reviews (reviewer #2 in particular). We do think that this paper should be in a general journal like eLife, partially because it is important that neurobiologists become at least familiar with the concepts and, to a limited extent, to the methods of control theory as practiced by engineers. Right now, the manuscript just won't work for most neurobiologists. But it can be written in a much clearer way. The import of the equations should be explained in the text and the derivations placed in a supplementary information section for specialists.

Initially, we went too far in avoiding being “tutorial”, but we now recognize that we missed the mark and that our approaches require far more explanation to be accessible to a general biology audience. We address the details of this Essential Revision in our detailed responses to the individual reviewers.

2) Might the model be tested directly/experimentally by perturbing the motor plant, for example weighting the fish by overfeeding or with a weight belt (reviewer #3)? There is a realization that these approaches might be unsuitable, but the experimenters are resourceful, and some perturbation might be found, e.g. clipping some of the ribbon fin (caudal quarter maybe).

The current paper uses computational models to test the idea that the controllers are robust to perturbations in the plant dynamics. However, as the reviewers suggested, one might directly test this idea on the animal itself by adding a weight jacket, overfeeding, or clipping the ribbon fin. Indeed, such experiments are described in Roth, Sponberg, and Cowan, 2014 (see p57, first column, last paragraph therein).

We have considered trying similar experiments over the last 10 years and have not yet found the silver bullet. First, adding weight to these fish is extremely hard for at least two reasons: (a) any added mass would have to be neutrally buoyant (such as a water balloon), limiting the materials that can be used, and (b) these are very delicate animals, and so suturing or otherwise attaching objects of significant mass to the animal is extremely challenging without potential injury to the animal. Second, we have considered more dramatic measures such as fin clipping, especially since animals often arrive to the lab with damage to their ribbon fins. However, in both of these cases, one would want to perform the mechanical alterations “on the fly” to avoid the longer-time scales of sensorimotor adaptation. Lastly, we are unaware of examples where our fish will eat a significant portion of their body mass in one feeding.

However, the issue raised by the reviewers is both fundamental and not well explained in the original manuscript. We have added a new discussion point on this subject (subsection “Feedback and Variability in Neural Systems”, fourth paragraph), where we point to studies that perform analogous plant perturbation experiments in other model organisms.

Reviewer #1:[…]Given this assumption and the authors prior papers, the experiments were well justified and well carried out. The analysis is, I expect, also well carried out. But here I think that the authors simply skipped too many steps in their derivations. This is the major problem that I think must be addressed in a revision.1) "We adopted the physics-based parametric model of locomotor dynamics of Eigenmannia described by Sefati et al., 2013", this is the 'controller dynamics'. This is Equation 1 in the Results and gives P(s). I think this is derived from Equation 4 is Sefati et al., 2013 (derived very nicely in the supplementary information). But, I fail to see the derivation. It is true that I did not expend a lot of effort trying to derive Equation 3 from Equation 4; however, I think that a typical reader should not have to make this attempt. The authors should make the derivation explicitly, perhaps in a supplementary information section (as was done in Sefati et al., 2013).

We have added the derivation and some clarifications to the Materials and methods. Specifically, we included a new subsection in Materials and methods, “Derivation of Plant Model”, which explains how P(s) is derived.

2) The authors then use a previous paper (Cowan and Fortune, 2007) to represent the behavioural response of the fish (Equation 2); this is well justified and gives G(s). The next point is the most critical and most novel. The authors then 'back calculate' the parameters for the controller, P(s), by using data driven behavioural model encapsulated in G(s). This results in Equation 3. At this point, Cowan and Fortune is cited and I got confused. Their closed loop transfer function, H(s), has an entirely different form from Equation 3. I am presumably missing something here and I think the authors really have to spell out in detail how the do the 'back calculation' and how it connects to Cowan and Fortune's results. Again, this would likely be a fairly detailed derivation and, as in the Sefati et al. paper, could be presented in supplementary information.

Yes, this was particularly confusing as some of the variables, including H(s), were renamed between previous publications and this manuscript. These differences have been resolved, and the essential components of the derivation added to the Materials and methods subsection “Inferring Controller using Plant and Closed-Loop Dynamics”.

Specifically, the closed-loop transfer function in the current manuscript, G(s) in Equation 2 is identical to H(s) in Equation 2 of Cowan and Fortune, 2007. Both manuscripts use the same approach: back-calculation of controllers from the closed-loop transfer function and plant dynamics. However, in the current manuscript we derived the controller for a general closed-loop transfer function G(s) and plant dynamics P(s) to obtain the solution given in Equation 3. This approach differs from Cowan and Fortune, 2007, where they solved for controllers for *hypothetical* (not experimentally estimated) plant dynamics. Nevertheless, although the final equations for the controllers look different, they originate from the same derivations.

3) Lastly, I was struck by the parallel between the authors' conclusions and that of Marder and colleagues. Dr. Marder and colleagues have shown, over many papers, that the output of the STG pattern generator can be produced by many different (with respect to parameters) STG models. There is no unique STG pattern generator and the average parameters don't match any real STG. A paper by O'Leary and Marder (2016, Current Biology) advances a similar argument to the one in this manuscript – that "The high variability in conductance densities in these neurons [9, 10] appears to contradict the possibility that robustness is achieved through precise tuning of key temperature-dependent processes."This paper uses Ca-dependent global feedback (Ca entry depends on activity) to maintain robust STG activity over a wide range of temperatures, despite the individual ion channels not being tuned for this temperature range. Although the systems are entirely different the principle seems, to me, to be the same. It would be great to add this to the Discussion.One overall conclusion of this paper is that "Physics-based approaches for modeling behaviors at lower levels (e.g., the spiking activity of all motor neurons) may lead to a very complex model that does not accurately capture high-level behavior." I fully agree and I also am convinced that the feedback tuning mechanisms of this manuscript and the O'Leary and Marder paper are the way to go for modelling complex systems – when one is dealing with even the electrosensory system of Eigenmannia, there are too many parameters to be estimated for a realistic approach to neural models. This is a powerful conclusion and giving it the wider context – crab STG and Eigenmannia electrosensory system and ribbon fin.

Thanks for suggesting the STG work. We are always interacting with people in the STG field, but somehow we didn’t make this connection. This is related to reviewer 2’s comments, and the two have been folded together in two new paragraphs at the end of the Discussion (subsection “Feedback and Variability in Neural Systems”, third and fourth paragraphs).

Reviewer #2:[…]This is a potentially highly impactful paper, however, I have three major concerns as listed below. Also, I should provide the caveat that I am neither an expert in models of fish locomotion nor in the kind of control-theoretic analyses used here. Therefore I must leave the technical evaluation of these aspects to other reviewers.1) Comprehensibility of this paper to a general audience of biologists. This is a highly technical paper which the authors are trying to squeeze into 4 main-text figures and the absolute minimum of explanatory text. Because of this choice, I think that this paper will not be comprehensible to most eLife readers (in its present state this seems much more appropriate for PLOS Comp Bio or some more specialized computational journal), thereby reducing its potential impact. I think the authors need to expand the text (and possibly include a couple of explanatory figures) to allow readers without control systems expertise to understand the paper. For example, a reader not already familiar with Nyquist or Bode plots will not be able to understand any of the key evidence for the authors' arguments presented in Figures 3 or 4. In the main text, the authors often cite their prior work rather than explaining essential components of their approach. This includes small things like not defining/explaining the variables in Equation 2 as well as explaining the key strengths of their model system – it took me several readings to understand what was meant by the nodal point providing a "linear proxy" for u(t).

We added subsections “Derivation of Plant Model” and “Inferring Controller using Plant and Closed-Loop Dynamics” to the Materials and methods that provide step-by-step derivations and applications of the equations.

In addition, we added and revised sections in the Materials and methods related to calculating variability in the model systems. These new subsections are “Bootstrap Estimate of Behavioral Variability” and “Model Variability.”

Other changes were made throughout to improve the clarity of the modeling and analysis; see for example, subsection “Estimating a Data-Driven Plant Model”.

We also reorganized the figures to improve comprehensibility of the paper as follows: the revised Figure 1 now shows the schematic of the system, a feedback diagram, and example data for ribbon-fin location. The revised Figure 2 now shows the plant model, including how model parameters were estimated. The revised Figure 3 focuses on the “computational brain transplant”. The revised Figure 4 presents the primary computational results on the role of feedback on variability. The Figure 5 in the current manuscript is identical to Figure 4 in the previous submission.

As background, we had two major concerns about describing the math when we were writing the original manuscript. First, we were concerned that too much emphasis on the equations would obscure the critical point that this work is driven by experimental data obtained in animals. That data is a central feature of the paper that is essential for all of the subsequent modeling and analyses. Second, we were concerned that the manuscript might become a form of tutorial on the application of control theory, which was not the primary objective of this manuscript.

That said, our long-term goal to help bring these concepts and approaches into the general lexicon of biologists who study movement and locomotion. In addressing the reviewers’ concerns, we think that the revised manuscript does a far better job of achieving that long-term goal while avoiding the pitfalls that concerned us before.

2) Dissociating the role of feedback control from that of plant dynamics. The authors present the study as attempting to distinguish between two alternatives – that behavioral robustness arises from either precise feedforward neural control or well-tuned feedback control. However it's not clear to me whether or how the authors have ruled out a third possibility – that the physics of the plant are such that behavioral output (forward-backward movement in the refuge task) is simply insensitive to variations in the motor command, at least over the range of motor command variations actually produced by the fish. In fact the result shown in Figure 3C would appear to be consistent with this possibility. The authors offer this interpretation:"Surprisingly, however, the variability remained below the trial-to-trial variability (Figure 3C, gray region versus dashed line). These results highlight the fact that sensory feedback can attenuate the output variability despite mismatch between the controller and plant pairs. In other words, feedback models do not require precise tuning between the controller and plant to achieve the low variability we observed in the behavioral performance of the animals."It seems to me that an equally plausible explanation is not that sensory feedback is compensating for "mismatch" errors, but rather that the physical plant is simply insensitive to errors. How can the authors rule this out? An analysis quantifying how much the natural variations in u(t) actually cause variations in y(t) would seem critical to answer this. It's possible that this issue is already implicitly addressed in the analyses provided, but if so I can't figure this out and I'd like the authors to clarify.

The reviewer is correct that the issue concerning the potential for the plant itself to be insensitive to errors was addressed implicitly in the original manuscript. Namely, if the plant were somehow able to reduce variability, it would manifest in the loop gain, since the loop gain is P(s)C(s). Indeed, the plants for the individual and merged fish are shown in Figure 2 and as can be seen, their gains vary by nearly a factor of 2 (from about 5 to about 10 at the lowest frequency). In addition, a related issue was mentioned in the original manuscript in describing parameter variability: “parameter estimates for the plant models varied two fold, but the parameter estimates for the closed-loop models varied around 15-20%“. Lastly, the plant gains are high (above 2 for low to mid frequencies) which means they will amplify errors. Indeed this is one of the features of robust feedback: open-loop sensitivity provides closed-loop robustness (see e.g. Csete and Doyle, 2002).

3) Expanding on a couple of discussion pointsThe authors conclude the Discussion as follows:"The dynamic synthesis of filters based on current sensory information may be a mechanism that shapes neural controllers on two time scales: for the moment-to-moment task dynamics and over the longer term for the maintenance of behavioral performances."I'd suggest that the authors expand on this idea a bit; at present it's not very clear to me how they think their simulation results (e.g. that the synthesis of the high-pass controller and low-pass plant yields a robust system) relate to the question of multiple timescales.

We agree that the current study in no way addresses longer time scales of adaptation and is only focused on moment-to-moment control. Thus, this discussion point was not well fleshed out, and we decided to eliminate it altogether.

Also I'd encourage the authors to expand on possible biological implementations of their findings, and speculate a bit on how those implementations might be similar or different from those known in other model systems. For instance, the controller described here is able to produce similar results (i.e., tracking the same stimulus albeit with greater variability) with different parameters (i.e., switched case) and is reminiscent of multifunctional neural circuits such as those characterized in the crustacean pyloric system. In addition, the characterization of how feedback reduces variability is similar to how the role of feedback and CPG entrainment is studied in multiple species including cats, stick insects, and crayfish. To be clear I'm not suggesting that the authors review all of these (or even any of them in great depth), but I'm sure readers will wonder how the authors theories relate to ideas from other systems.

We agree this is an interesting connection, and is related to reviewer 1’s comments, which we address in tandem by adding two paragraphs to the end of the Discussion (subsection “Feedback and Variability in Neural Systems”, third and fourth paragraphs). Specifically, the work in stick insects includes perturbations to the mechanical plant (angle of the walking surface) with no changes in the kinematics. This is akin to the suggestion above for new experiments that perturb the animal’s plant.

Reviewer #3:I've been thinking a lot about this paper and while I argue that it should eventually be published in eLife, I think (for this broad readership journal) it could use some expansion regarding pedagogy on the control aspects. I find it very challenging to decide if I believe that the "simulated swapping" tells me anything fundamental about the system, or if it is just a computational exercise. I think it is not, but it is hard for me to evaluate given all the assumptions going into "gray box" model. What if the black box model isn’t the correct one? In particular, since the authors likely have room in terms of word count, and certainly have space in terms of numbers of figures, I think a bit more explanation of the methods to demystify the story would be helpful.

We have expanded our description of the modeling and computational methods. See response to reviewer #2, issue #1.

Based on the first paragraph of the Introduction, I would have expected to see how the controller changes when fish lose or gain weight (or some manipulation). But I am not convinced by their statement "These changes likely result in variable mismatch between the neural controller and the locomotor plant of individual animals. This mismatch is similar to that induced by swapping plants and controllers across individuals, suggesting that moment-to-moment variability can also be eliminated through sensory feedback." Is this true? Why not do the (obviously harder) experiment of indeed letting the fish eat a lot (or not) and measuring the controller. That way, the authors don't have to do a virtual swap? Even better, why not add weighted "jackets" to the fish as a real "plant" manipulation? I think if they could show this, the paper would become outstanding (instead of "just" very cool as it is now).

We have thought carefully about the types of experiments that we could attempt – including feeding, adding neutral buoyancy ballast, fin clips, and virtual reality to modulate feedback. Ultimately we want to make neurophysiological recordings from the neurons that comprise the neural controller in freely swimming fish that are performing tracking while systematically perturbing the locomotor plant.

To address this concern, we have added references and discussion about other species in which these sorts of manipulations have been made. See subsection “Feedback and Variability in Neural Systems”, third and fourth paragraphs.

This phrase in the Abstract "virtually indistinguishable between conspecifics" is a bit overwrought. What does "virtually indistinguishable" mean?

This imprecise statement has been replaced.